# Minimizing Chronic Kidney Disease (CKD) Underdiagnosis Using Machine Learning

**Lawrence Huang, B.S.[1], Sachin Shankar, M.S.[2], Keyvon Rashidi, B.S.E.[3], Dany Alkurdi, A.B.[4], Felipe Giuste, Ph.D.[5]**

[1]Warren Alpert Medical School of Brown University
[2]University of Cincinnati College of Medicine
[3]Texas A&M School of Engineering Medicine
[4]Icahn School of Medicine at Mount Sinai
[5]Emory University School of Medicine

lawrence_huang1@brown.edu, shankasc@mail.uc.edu, keyvon@tamu.edu, dany.alkurdi@icahn.mssm.edu, felipe.giuste@emory.edu

## Abstract

Chronic Kidney Disease (CKD) is a prevalent and devastating progressive disease affecting up to 14% (>35.5 million individuals) of the United States population and costing Medicare well over $64 billion annually. As many as 90% of individuals with CKD are undiagnosed, indicating the need for better tools to diagnose CKD and prevent unnoticed disease progression. However, current methods of assessing CKD have limitations regarding accessibility, practicality, and accuracy. This study seeks to address these limitations by developing a data-driven method to assess CKD risk from a large open-source database of electronic health records that has not previously been applied for CKD prediction. Machine Learning (ML) methods were used to develop a software tool to predict patient CKD status with patient-specific demographic data, vital signs, and past medical history. Of the ML models used in this study, a Random Forest Classifier had the best performance in predicting CKD diagnosis correctly with an accuracy of 0.875, an Area Under the Receiver Operating Characteristic Curve of 0.927, and an F1 score of 0.765. Our results indicate that ML-based approaches can help facilitate early screening and intervention for patients at risk of CKD. For progressive diseases like CKD that become more devastating and expensive to treat as they progress, high rates of missed diagnoses can be reduced by ML models leveraging electronic health record data.

## Introduction

### Disease State Fundamentals

Chronic Kidney Disease (CKD) is a disease that involves the loss of renal function over time. In CKD, there is a progressive decline in the glomerular filtration rate (GFR), leading to the accumulation of toxic waste products in the body. Pathophysiologically, CKD can result from direct damage by diseases such as diabetes, hypertension, or systemic autoimmune diseases (Vaidya and Aeddula 2024).

Currently, in the United States, the prevalence of CKD is approximately 35.5 million (14% of the US population or 1 in 7 individuals in the US) (System 2023). Assuming a stable CKD prevalence (14%) and a US population increase of 0.5% in 2023 (around 340 million), the estimated incidence of CKD is 1.57 million. Additionally, CKD prevalence increases with age, affecting 34% of individuals aged 65 and older, compared to 14% of those aged 45-64, and 6% of individuals aged 18-44 (Centers for Disease Control and Prevention 2023). Moreover, CKD occurs with a higher prevalence in women (14%) compared to men (12%) (Centers for Disease Control and Prevention 2023; System 2023). CKD is often a silent disease, with as high as 90% of individuals going undiagnosed (Centers for Disease Control and Prevention 2023). With the ever-aging US population and increasing prevalence of hypertension and diabetes, the leading causes and associated conditions with CKD, an increasing prevalence of CKD is to be expected (Rossing et al. 2022). Concerning CKD mortality, the adjusted rates are approximately 98.5 per thousand person-years for individuals aged 66 years and older, making and finding advancements in the diagnosis, monitoring, and treatment of CKD a critical concern (System 2023).

## Previous Work and Study Scope

Traditional solutions monitoring CKD disease progression traditionally require physiological or lab-based monitoring of GFR.

A technique known as estimated GFR (eGFR) uses age, serum creatinine levels, and gender to predict GFR based on average population measurements. As such, it can be rapidly assessed in the clinic or emergency room, making kidney screening and monitoring relatively practical over time. However, as the eGFR relies on the serum creatinine concentration, it is subject to several confounding variables, namely, muscle mass and diet. Alternatively, measured GFR (mGFR) is a more accurate technique that requires time-consuming and expensive serial measurements; it is only available in specialized clinics and research facilities. Finally, at-home urine test strips can detect the presence of abnormal concentrations of proteins in the urine. However, such results are qualitative, and established quantitative metrics such as urine albumin-to-creatinine ratio (UACR) cannot be calculated using such test strips (Folkerts et al. 2021).

Previous work has applied Machine Learning to the assessment and diagnosis of CKD (Krisanapan et al. 2023; Qezelbash-Chamak et al. 2022). Both supervised and unsupervised machine learning methods have been applied to predict CKD with classifier algorithms such as XgBoost, Support Vector Machines, and Decision Trees (Islam, Majumder, and Hussein 2023; Debal and Sitote 2022; Bai et al. 2022; Qin et al. 2020; Khalid et al. 2023; Dashtban et al. 2023; Ilyas et al. 2021).

This study aims to further advance the development of automated and efficient diagnosis of CKD patients through utilization of the MIMIC-IV database (Goldberger et al. 2000; Johnson et al. 2023b,a). Due to the aforementioned limitations in traditional methods of diagnosing and monitoring CKD, this study seeks a data-driven method of early CKD prediction and diagnosis to improve the quality of patient care.

### Clinical and Economic Significance

Concerning Medicare spending, CKD accounts for a significant portion of healthcare costs, totaling over $64 billion, greater than 20% of Medicare's budget, in 2015; such numbers have increased significantly over the years as the incidence of CKD has increased (Liu and Zhao 2018). Additionally, almost 90% of CKD patients who advance to End-Stage Renal Disease (ESRD) undergo hemodialysis, a treatment that costs Medicare an average of more than $80,000 annually per patient (Liu and Zhao 2018). Analyses have shown that early CKD screening could achieve an incremental cost-effectiveness ratio (ICER) as low as $6,342 per Quality-Adjusted Life Year (QALY) (Yarnoff et al. 2017). ICER is a measure of the cost to provide a patient with one additional year of life, standardized for certain quality-of-life measures. This would represent substantial potential savings compared to the exorbitant costs of treating advanced-stage CKD, which often exceed $80,000 per patient annually for treatments like hemodialysis (Liu and Zhao 2018; Yarnoff et al. 2017).

## Methods

### Data Acquisition and Preparation

This study utilized the MIMIC-IV dataset, a large-scale deidentified collection of hospital EHR data grouped by a patient stay from a single large tertiary academic medical center (Johnson et al. 2023b,a; Goldberger et al. 2000). To our knowledge, this database has not been previously utilized for CKD prediction. A patient stay in this database was defined as each distinct encounter for a given patient within the health system. The data was originally in CSV form, which was converted into a PostgreSQL database, enabling the extraction and querying of relevant tables in a relational format. Key tables extracted included demographic information, vitals, and diagnostic records containing ICD codes. For our model, each observation consisted of the demographic data (age, gender), vitals (max body mass index, maximum systolic blood pressure, minimum diastolic blood pressure), and ICD codes for a given patient during a given stay. All of our code was made publicly available

to facilitate reproducibility and future work (https://github.com/lawrenceh1850/Huang-et-al-AAAI-CKD).

### Determination of CKD Status

CKD status was determined by identifying specific ICD codes associated with CKD. All observations with a CKD code in this set were deemed CKD observations. All other observations were designated non-CKD patients.

### Feature Selection

To identify the most relevant features that could be used for predicting CKD amongst non-CKD patients, this study focused on the prevalence of non-CKD ICD codes in patients diagnosed with CKD. This prevalence-based feature selection approach aimed to determine a robust set of features significantly associated with CKD.

After initially selecting all non-CKD ICD codes that were observed in more than 5% of the CKD observations, a correlation of each feature with CKD status was performed. This resulted in the identification of a few highly correlated ICD codes, such as the presence of dialysis, that were then filtered out so the model had to learn to predict potential CKD status from only non-CKD associated ICD codes. Our cleaned dataset also resulted in 99,922 unique CKD observations and 265,771 unique non-CKD observations. The final number of observations was 365,693.

### Data Preprocessing

The selected data underwent preprocessing, which included converting ICD codes into a one-hot encoded format. This step transformed categorical data into a binary matrix, facilitating its use in machine learning models. Missingness analysis showed that no features were missing in more than 75% of patients, and no observations were missing more than 40% of features. Missing data was filled in with a k-nearest neighbors imputation method using the 3 closest neighbors.

The dataset was then divided into training (n=292,554) and test (n=73,139) sets. Comparison of features across both train and test sets showed high degrees of matching prior to model evaluation, as shown in Table 1.

|  | Train Set | Test Set |
|---|---|---|
| Number of Patients | 292,554 | 73,139 |
| Features |  |  |
| Age in Years | 60.07 ± 17.86 | 59.98 ± 17.88 |
| Sex (M / F) | 48.79% / 51.21% | 48.85% / 51.15% |
| Max Systolic BP | 134.16 ± 33.07 | 134.12 ± 33.07 |
| Min Diastolic BP | 74.51 ± 16.23 | 74.58 ± 16.23 |
| CKD Stages |  |  |
| Stage 0 | 212,616 (89.76%) | 53,155 (89.61%) |
| Stage 3 | 11,284 (4.76%) | 2,864 (4.83%) |
| Stage 4 | 4,110 (1.74%) | 1,035 (1.74%) |
| Stage 5 | 8,871 (3.74%) | 2,261 (3.81%) |

Table 1: Comparison of Characteristics between Train and Test Sets. All continuous variables show mean ± standard deviation. BP = blood pressure

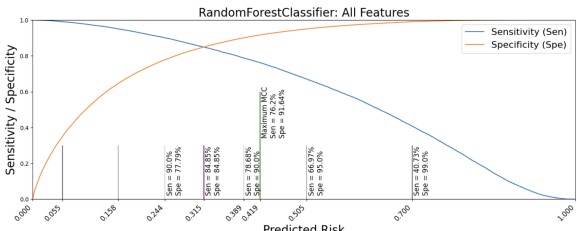

Figure 1: Random Forest Classifier Performance at a range of classification thresholds

## Model Training and Validation

Three different machine learning models were trained on the training dataset to predict CKD status. These models included Logistic Regression (LR), k-Nearest Neighbors (kNN), and Random Forest (RF). Hyperparameter tuning was done in this step using cross-validation. The best-performing hyperparameters for each model type were selected, and the final three models proceeded to model evaluation.

## Model Evaluation

The relative performance of each model architecture was evaluated using the Area Under the Receiver Operating Characteristic (AUROC) curve. The model threshold that best balanced sensitivity and specificity for each model architecture was selected using the Matthews Correlation Coefficient (MCC). The MCC is an alternative model statistical measure to more well-known scores, such as F1, that optimizes performance across all four categories of the confusion matrix. This measure is particularly reliable because it considers the balance between the size of the dataset's positive and negative elements, ensuring the proportionate representation of all categories in the evaluation (Chicco and Jurman 2020).

## Results and Analysis

A comparison of the performance of the three machine learning models is seen in Table 2. LR corresponds to the performance of the logistic regression classifier, RF is random forest, and kNN is k Nearest Neighbors.

| Model | AUROC | Accuracy | Precision | Recall | F1 |
|---|---|---|---|---|---|
| LR | 0.8342 | 0.8126 | 0.6726 | 0.6124 | 0.6411 |
| **RF** | **0.9266** | **0.8753** | **0.7877** | **0.7440** | **0.7653** |
| kNN | 0.8697 | 0.8289 | 0.7814 | 0.5190 | 0.6237 |

Table 2: Comparative Performance Metrics of Logistic Regression (LR), Random Forest (RF), and k Nearest Neighbors (kNN) Models

The Random Forest classifier was selected as our final model architecture, as it had the highest AUROC across all models. MCC was then used to select the best decision threshold for our trained RF model for the best combination of accuracy, precision, and recall.

Figure 1 shows model performance MCC at various thresholds (corresponding to each of the vertical black lines). The tallest green line shows the maximized MCC, denoting the prediction threshold that our model selection process picked. In other words, given a model-determined probability greater than 41.9%, the Random Forest Classifier would predict a given patient as CKD-positive. Note that the sensitivity and specificity corresponding to maximized MCC in Figure 1 are slightly different from our test sensitivity and specificity in Table 3. This is because the MCC was calculated for the model on the training set, a threshold was set, and then this threshold was evaluated on the testing set.

| | Predicted Negative | Predicted Positive | Total |
|---|---|---|---|
| CKD Negative | 243,553 (66.60%) | 22,218 (6.08%) | 265,771 |
| CKD Positive | 25,580 (6.99%) | 74,342 (20.33%) | 99,922 |
| Total | 269,133 | 96,560 | 365,693 |

Table 3: Confusion Matrix for the Random Forest Model

The confusion matrix (Table 3) shows the percentages as proportions of the total number of observations. Positive predictive value (PPV) was 77% and negative predictive value (NPV) was 91%. The PPV calculates the conditional probability of the patient having CKD given that the model predicted they had CKD, whereas the NPV shows the conditional probability of the patient not having CKD given that the model predicted they did not have CKD.

## Discussion

The results presented indicate that the Random Forest Classifier is a robust model for identifying CKD in patients, with implications for Value-Based Care (VBC) for CKD management.

### Predicting Undiagnosed CKD Patients

Identifying individuals with undiagnosed CKD or at increased risk is essential to prevent progression to end-stage kidney failure. Our early prediction model aimed to identify high-risk individuals before they reach end-stage renal failure (ESRD). At the maximum MCC, the model's high specificity (91.64%) and reasonable sensitivity (74.4%) underscore its potential for early detection of CKD. The model was selected for its high specificity, as high specificity indicates relatively fewer false negatives and prioritizes avoiding missing diagnoses. This targeted approach ensures sick patients are treated appropriately and with early interventions.

The lower sensitivity is an optimal tradeoff in this case, as more false positives are identified. Examining these cases

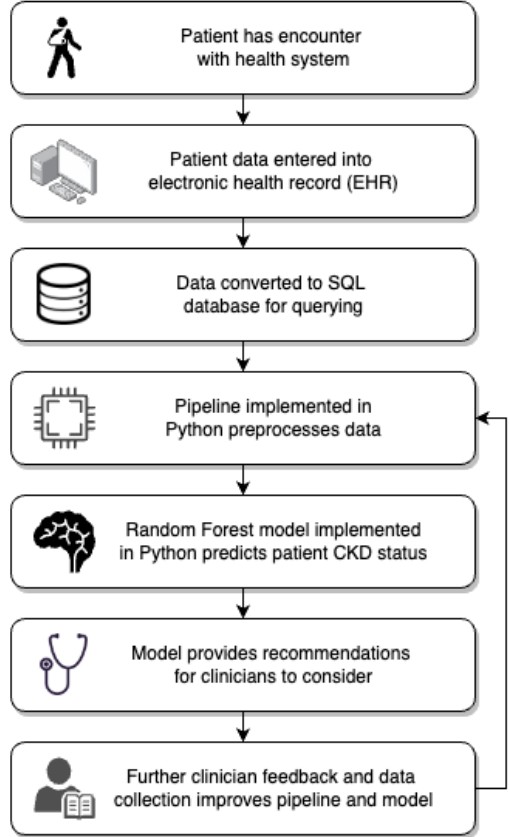

Figure 2: Envisioned integration of CKD model into clinical workflow

more closely could reveal a subset of patients at an increased risk of developing CKD, as non-CKD ICD codes found amongst CKD patients could be potential covariates for predicting CKD in non-CKD patients. With further work and prospective studies, those false-positive patients could be prospectively studied and determined to indeed be at greater risk of developing CKD.

## Value-Based Care in Chronic Kidney Disease

Value-based care (VBC) in CKD management prioritizes high-quality care while managing costs. VBC measures value by the quality of care relative to cost. Models like the ESRD Treatment Choices and the Kidney Care Choices, part of the Advancing American Kidney Health initiative, embody this by incentivizing improved care for kidney disease at lower costs. Key aspects of VBC in CKD include adjustable visit frequencies for patient stability, diligent monitoring to maintain kidney function, and proactive vascular access planning (Brady et al. 2019).

Our predictive model aligns with these VBC goals by potentially reducing costs while improving quality through early CKD detection, targeted patient screening, and appropriate follow-up. For example, a digital diagnostic tool like the one developed in this study would intersect well with existing payer population health programs. The algorithm could be deployed via a payer's mobile app and then automatically provide options to provide referral to lab testing, provider visits, or emerging home testing kits.

It would be ideal to deploy this algorithm interoperably with EHR systems as a SMART on FHIR app to enable clinicians across different systems access to the tool, facilitate point-of-care usage of the app for clinical decision support, and provide the EHR data required for continuous model refinement.

## Limitations

Any ML model must be tested and validated in diverse real-world settings before deployment in clinical use cases. The performance of the model might differ based on the population demographics, comorbidities, and healthcare access, which can influence CKD prevalence and presentation. Regular calibration of the model with up-to-date patient data will ensure it remains relevant and accurate over time, further aligning with VBC goals by continuously improving patient care standards. Our relatively lower sensitivity and positive predictive value show the difficulty of the model in identifying CKD from non-CKD ICD codes. This accords well with clinical difficulty in diagnosing CKD.

Further work could identify which ICD codes specifically were identified by the models as being predictive of undiagnosed CKD, and clinical testing for those patients could further validate this status, as seen in Figure 2. With further study, this would be possible as the ML models chosen in this study are interpretable. Additionally, techniques to calibrate the model output probabilities to more closely match the actual clinical probability of CKD for a given patient could be explored further. Without calibration, probabilities outputted by the model as degrees of belief as to whether a patient has CKD cannot be directly interpreted. Performing calibration would make the model even more useful as patients could be more definitively screened with borderline classification probabilities for CKD.

Especially important in the area of CKD, it is well known that the use of race in eGFR calculations in the past led to inequitable treatments (Uppal et al. 2022). As such, analyses should be performed by comparing model predictions between socioeconomic, racial, and ethnic groups to ensure that our algorithm is not propagating any biases in the data. Any identified biases would have to be corrected using bias mitigation methods developed specifically for machine learning models.

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
