# OpenReview forum: "Minimizing Chronic Kidney Disease (CKD) Underdiagnosis Using Machine Learning"
_AAAI.org/2024/Spring_Symposium_Series/Clinical_FMs — AAAI 2024 SSS on Clinical FMs_

### Official Review · Reviewer_42Yx · 2024-02-14

**Rating:** 7
**Confidence:** 3

**Review:**

The paper discusses machine learning for chronic kidney disease prediction. Multiple baselines are tested. The strength of the paper includes the detailed background introduction and problem-driven study, with thorough data analysis and experiment discussion. The code is made available online. The weakness of the paper includes too few baselines being tested, not standardized and lacking comparison to some foundation models (that are not specifically for chronic kidney disease but can be easily tailored to do so). Figure 1 is too small, especially with too small fonts. The Discussion section lacks more tables/figures/statistics to back up the sentences.

---

### Official Review · Reviewer_sYoX · 2024-02-14
**A valuable approach but not directly related to the conference main theme**

**Rating:** 3
**Confidence:** 3

**Review:**

The paper addresses a crucial healthcare challenge with interesting objectives. It leverages machine learning techniques to improve the identification and management of CKD, which is a significant contribution given the global burden of the disease. The methodology employed is robust, and the results presented indicate a high level of effectiveness in achieving the paper's goals.

However, despite the promising application and outcomes, this study primarily utilizes classical machine learning approaches without incorporating any pre-trained models, which diverges from the conference's emphasis on innovative model foundations and applications in clinical settings. This discrepancy could be seen as a deviation from the core subject matter expected at the conference.

Furthermore, while the paper effectively demonstrates the performance of ML techniques in reducing CKD underdiagnosis, it falls short in situating its findings within the broader context of current state-of-the-art methods. A comparative analysis, not limited to data-driven approaches, in terms of precision and cost-effectiveness, would have greatly enriched the paper. Such a comparison is essential for understanding the true value and innovation of the proposed method over existing strategies.

In conclusion, while the application of machine learning techniques to tackle CKD underdiagnosis is indeed valuable, the paper's methodological approach lacks the novelty and direct relevance to the "Clinical Foundational Model" conference theme. The absence of a comparative analysis with state-of-the-art methods further limits the paper's contribution to the field. Therefore, despite its potential impact in healthcare, the paper may not meet the innovation threshold required for acceptance into the conference.

Pros:
- Tackling a challenging problem in healthcare
- Great accuracy

Cons:
- Not related to the main conference theme
- Lack comparison with SoTA methods
- Lack a cost reduction analysis

---

### Official Review · Reviewer_MKZD · 2024-02-20
**Comparison of Random Forest, Logistic Regression, and KNN models for CKD prediction on MIMIC-IV Dataset**

**Rating:** 6
**Confidence:** 4

**Review:**

## Summary
This manuscript explores the application of traditional machine learning models (Random Forest, Logistic Regression, and kNN) to the prediction of Chronic Kidney Disease (CKD) on the MIMIC-IV dataset using CKD-related ICD codes as labels and non-CKD-related ICD codes and demographic variables as features.  This is an important healthcare problem and establishes a baseline on MIMIC-IV dataset.  In general, the methodology for data sampling, splits, evaluation metrics are well done.  This manuscript would be stronger with the inclusion of more advanced models such as gradient boosted trees which often achieve stronger performance in many healthcare prediction tasks with tabular data.

 The authors are somewhat vague in the real-world application of this model--Is the goal to create a clinical decision support/forecasting model while the patient is still in the hospital? Is the goal to create a model that can detect if we forgot to mark the presence of CKD in an encounter for billing after the ICU stay?  This is not clearly stated and these are different questions that require different modeling approaches. The design of the presented models are mainly useful for the second line of questioning.  However, in an ICU population where creatinine and eGFR values are very frequently measured, why not compute a clinical baseline using traditional diagnostic definitions for comparison? This is currently absent from the manuscript and inclusion would make the research much stronger. More on this below.

## Pros
* Important healthcare problem. Research establishes baseline for CKD prediction task from ICD + demographic data on MIMIC-IV dataset.
* Well designed experiments, train/test split
* Large sample size & statistically robust
* Good choice of evaluation metrics (AUROC, AUPRC) and threshold selection rationale with MCC

## Cons
* I consider Random Forest to be a simple model. The findings would be more interesting if a more advanced model such as Gradient Boosted Trees was also compared, especially since Gradient Boosted Trees often achieve state-of-the-art performance on many healthcare prediction tasks using tabular features.
* Poor word choice in second paragraph of "Predicting Undiagnosed CKD Patients" section.  The authors write "The lower sensitivity is a benefit in this case...", but lower sensitivity is never a benefit because we always desire higher sensitivity and specificity.  I think the authors mean that it is an optimal trade-off.
* This analysis pools all CKD classes together rather than predicting individual CKD I, II, etc.  It would be more interesting/valuable if authors designed experiments to predict individual CKD classes in addition to predicting presence/absence of CKD.  This is because management and healthcare cost of CKD I/II patient (often no dialysis, medication management) is very different than CKD IV/V (patients usually more ill, more comorbidities, and/or require dialysis).  The utility of this prediction model would be significantly improved if models were able to predict CKD classes.
* MIMIC IV dataset is derived from real electronic health records of ICU patients where there is a temporal nature to the data. Some diagnoses & ICD codes may be present in certain days/encounters earlier than others. The authors' methods indicate that ICD codes extracted correspond to all ICD codes for a given stay--that is at time of discharge. Choosing this time point limits the utility of this predictive model as many diagnoses (ICD codes) will be accumulated during the ICU stay and may not be present on admission.  The value of this model thus becomes post-encounter detection (e.g. CKD billing code is missing), not for prospective clinical decision support. A more clinically useful time point for prediction for diagnostic/clinical decision support would be earlier in the hospital stay (e.g. day 1 or day 2 of admission).  The target use case of the proposed model should be more clearly stated; currently it is vague. The chosen use case for this model would then inform different choices for data selection and model development.
- Since the study population are ICU patients, I expect almost all to have multiple creatinine value or eGFR determined in their laboratory studies. The creatinine and eGFR values are how CKD is diagnosed using diagnostic criteria such as KDIGO.  The authors tangentially acknowledge these traditional diagnostic criteria in "Previous Work and Study Scope" section, but do not actually compute these values as a clinical baseline.  The proposed line of research would be much stronger and clinically useful if authors determined CKD from traditional diagnostic criteria (which is still widely used in clinical medicine) as a baseline for comparison, or used the computed values as ground truth instead of ICD codes. This should be possible for most patients in the MIMIC dataset because of the high availability of creatinine and eGFR data in ICU patients. Currently authors compare against presence of ICD codes that are related to CKD, but this ground truth may be inaccurate given that ICD codes are primarily used for billing purposes.  In theory ICD codes should reflect the patient's clinical reality but in reality it may not necessarily be reflective since it requires billing staff or healthcare staff to denote the presence of the diagnosis in the patient's EHR. Thus relying on ICD codes as ground truth may actually lead to your model to under-diagnose CKD.

---

### Official Review · Reviewer_eh7c · 2024-02-24
**Minimizing Chronic Kidney Disease (CKD) Underdiagnosis Using Machine Learning**

**Rating:** 9
**Confidence:** 5

**Review:**

The study presents a compelling approach to enhancing the diagnosis of Chronic Kidney Disease (CKD) using a data-driven method. By utilizing the MIMIC IV, a large, open-source database of electronic health records (EHRs), the research employs ML techniques to determine best approach to assess the risk of CKD.

The methodology is comprehensive, incorporating patient-specific demographic data, vital signs, and past medical history to predict CKD status accurately. The authors meticulously detail their data sources, the process of pre-processing, and the strategies employed to handle missing data, showcasing a robust methodological framework. The training and test datasets are balanced across CKD disease stages, which is crucial for the reliability of the predictive models. The authors statistical analysis is sound and supported by clear tables and figures.

They show that the Random Forest Classifier was identified to have the best performance, achieving an accuracy (0.875), an  Area Under the Receiver Operating Characteristic Curve (0.927), and a F-1 score (0.765). These results supports authors conclusion that the  random forest CDK classifier algorithm maybe effective in identifying patients at risk of CKD, particularly those who may be under-diagnosed due to health disparities.

While the study is not about foundation models, the authors demonstrate and remind us of that simple algorithm maybe sufficient to solve pervasive healthcare problems.